# Death and dying in prehospital care: what are the experiences and issues for prehospital practitioners, families and bystanders? A scoping review

Michelle Myall ![ORCID],[1] Alison Rowsell,[1] Susi Lund,[1] Joanne Turnbull,[1] Mick Arber,[2] Robert Crouch,[1,3] Helen Pocock,[4,5] Charles Deakin,[4,6] Alison Richardson[1,3]

For numbered affiliations see end of article.

**Correspondence to**
Dr Michelle Myall;
M.Myall@soton.ac.uk

## ABSTRACT

**Objective** To identify the factors that shape and characterise experiences of prehospital practitioners (PHPs), families and bystanders in the context of death and dying outside of the hospital environment where PHPs respond.

**Design** A scoping review using Arksey and O'Malley's five-stage framework. Papers were analysed using thematic analysis.

**Data sources** MEDLINE; Embase; CINAHL; Scopus; Social Sciences Citation Index (Web of Science), ProQuest Dissertations & Theses A&I (Proquest), Health Technology Assessment database; PsycINFO; Grey Literature Report and PapersFirst were searched from January 2000 to May 2019.

**Eligibility criteria for selecting studies** Qualitative and mixed methods studies reporting the experiences of PHPs, families and bystanders of death and dying in prehospital settings as a result of natural causes, trauma, suicide and homicide, >18 years of age, in Europe, USA, Canada, Australia and New Zealand.

**Results** Searches identified 15 352 papers of which 51 met the inclusion criteria. The review found substantial evidence of PHP experiences, except call handlers, and papers reporting family and bystander experiences were limited. PHP work was varied and complex, while confident in clinical work, they felt less equipped to deal with the emotion work, especially with an increasing role in palliative and end-of-life care. Families and bystanders reported generally positive experiences but their support needs were rarely explored.

**Conclusions** To the best of our knowledge this is the first review that explores the experiences of PHPs, families and bystanders. An important outcome is identifying current gaps in knowledge where further empirical research is needed. The paucity of evidence suggested by this review on call handlers, families and bystanders presents opportunities to investigate their experiences in greater depth. Further research to address the current knowledge gaps will be important to inform future policy and practice.

## Strengths and limitations of this study

► To our knowledge this is the first review to focus on practitioner, family and bystander experience of death and dying in prehospital care.

► The review will help to make better sense of prehospital practitioners, families and bystander experiences and to prioritise, from these perspectives, ways to improve these experiences through support and training that includes ethical issues and challenges faced by the work of death and dying.

► The search strategy aimed to balance sensitivity and precision, and pragmatic decisions helped achieve this balance to target studies most likely to be relevant. However, these decisions may also have increased the risk of missing relevant records.

► A focus on healthcare systems similar to that of the UK was intended to increase transferability of findings. However, differences between these systems, and exclusion of healthcare systems that did not parallel the UK's may limit transferability.

## INTRODUCTION

Prehospital care (PHC) is an essential part of the emergency and urgent care continuum in contemporary healthcare systems across the world[1 2] and includes urgent and emergency medical care that patients receive outside of the hospital setting. In the UK, PHC is provided primarily by regionally based National Health Service (NHS) Ambulance Trusts and comprises other services such as patient transport and NHS 111, a 24-hour online and telephone urgent care service.[3] In some UK regions services are delivered by other providers, including charities and private companies, but in line with NHS principles remain free at the point of access, with some limited exceptions in England. For example, it is a requirement for some overseas students to pay an NHS surcharge or take out private health cover. However, the system in the UK is not reflected worldwide. In Australia, other than in Queensland and Tasmania, ambulance services are covered either by private health insurance or are out of pocket, unless an individual is eligible for

a concession such as those over 65 years of age, where cover is free, or in some states offered at a reduced rate.[4] Ambulance services in the USA are operated by private (for-profit and non-profit) and public entities, and with no free universal healthcare in the USA are typically paid for by private health insurance or federally funded programmes such as Medicare and Medicaid.[5] Evidence suggests the nature of ambulance provision can influence the service provided.[6]

In the UK there is an increasing demand for PHC services.[3] [4] In England for the period 2014–2015 the number of emergency 999 calls to ambulance switch-boards totalled 9 million, a rise of over 500 000 from the previous year[7] an increase mirrored in other healthcare systems.[8] [9] Factors contributing to increasing demand are complex and reflect the extent to which work carried out in PHC has been required to react to changes in other parts of the healthcare system.[10] For example, ageing populations with complex needs,[11] [12] difficulties in accessing general practitioner (also known as primary care physician) services,[13] and greater patient expecta-tions and how patients seek help[14] have meant the remit of the service has changed.[15] As an evolving service prehos-pital practitioners' (PHP) work has become increasingly varied and complex requiring a clinically trained work-force comprising a variety of staffing combinations, such as medical dispatchers, doctors, paramedics, emergency medical technicians and emergency care practitioners.

The nature of the work of PHPs can range from responding to time-critical emergencies for serious or life-threatening injuries or illnesses, such as cardiac arrests, calls that are less immediately time critical but still serious, to those not considered life threatening. Out-of-hospital cardiac arrest (OHCA) is a lead cause of death in industrialised society,[16] [17] and while it represents a small proportion of PHC (approximately 0.5% of calls to English ambulance services) national response targets place a significant strain on services. In England, during 2014, the ambulance service treated 28 729 cases of OHCA alone,[18] and evidence suggests numbers rising year on year.[19] It is also important to note that of the 60 000 OHCA calls attended by ambulance personnel, where treatment was not appropriate, patient assessment, breaking bad news and supporting family and bystanders were also an essential part of care delivered. Increasingly, the ambulance service also responds to calls for people who have life-limiting conditions and approaching the end of life (EoL).[20]

As first responders to crises that immediately precede death, confronting death and dying is an intrinsic part of the work of PHPs.[21] When providing care to someone who is dying, or dies, they are often required to make challenging decisions and deal with the clinical and emotional aspects of a situation simultaneously.[22] This requires management of their own feelings and responses, as well as others present including family and bystanders who may or may not have an established relationship with the person. In some countries PHPs can verify death.

For example, in the UK since 2004, Recognition of Life Extinct guidelines permit qualified PHPs to confirm death and cease resuscitation in the field.[23] Similarly, in the USA and Canada paramedics can confirm death on scene.[24] [25] This aspect of their work has the accompanying responsibility of informing relatives or others present that the patient has died.

The provision of care to someone who is dying, and their relatives, has been identified as one of the most stressful human experiences[22] [26] and acknowledged as challenging for healthcare professionals (HCP) gener-ally.[27] [28] For those working in PHC, this may be more complex partly because ambulance crews are immedi-ately required to assess and respond to a situation,[26] and the transient nature of the job may not present oppor-tunities to establish rapport with a dying patient or their relatives, which may have a lasting impact for survivor families into bereavement. In dealing with death and dying PHPs engage in 'moral work' needed to navigate the difficult ethical challenges they experience in this particular context. These include managing expecta-tions from families and providing patient-centred care that is in the best interests of the patient. The recent global COVID-19 pandemic has posed additional ethical demands on PHPs.[29] In particular, decision-making may be influenced by the rationing of scarce resources, being unable to provide the level of individual care to patients and families they would under more usual conditions, not being deployed to patients they would have attended previously and leaving patients at home who otherwise would have been transferred.[30] [31] Such ethical challenges may lead to increasing levels of moral distress[32] [33] for PHPs who are unable to pursue what they consider to be the right course of action due to varying internal and/or external constraints.

To date, little is known about the effects on families and others present, or support mechanisms in place for rela-tives or bystanders who witness an incident when a patient dies where PHPs attend. Similarly, while we know that PHPs encounter death and dying routinely in the course of their work, we understand less about the nature of this aspect of their job, the impact of dealing with death and dying, how they manage situations where a patient dies especially when they do not always have opportuni-ties to debrief with their colleagues,[34] or their emotional, psychological, educational and training needs in relation to death and dying.

This review is needed to better inform our under-standing about the experiences and needs of PHPs, fami-lies and bystanders in the prehospital context of death and dying in order to identify areas for further research.

### Aim of the review
In this paper we present a scoping review that explores evidence guided by the question: *What is known about the factors that shape and characterise experiences of PHPs, families and bystanders in the context of death and dying outside of the hospital environment where PHPs respond?*

**Table 1** PEO framework

| P | Population and problems | Family members/bystanders/witnesses/healthcare professionals who have experience of being present when a patient is dying or dies when responded to by prehospital services. |
|---|---|---|
| E/I | Exposure/issue | Death and dying where prehospital services respond. |
| O | Outcome/themes | Experiences and views of death and dying where prehospital services respond. |

Specific objectives were to:

1. Ascertain experiences of PHPs when providing care to patients, and supporting families and bystanders, and their own support and training needs.
2. Understand types of PHPs' behaviours and communication strategies enacted for family members and bystanders.
3. Explore families and bystanders' experiences and needs and identify any existing support mechanisms.

## DESIGN AND METHODS

A scoping review methodology[35–37] was selected as the most appropriate for systematically mapping the literature and identifying key themes, concepts and gaps in knowledge. We used Arksey and O'Malley's[38] five-stage framework for scoping reviews which includes identifying the research question, identifying relevant literature, selection, charting the data and collating, summarising and reporting the results. We also used Levac *et al*'s recommendations to strengthen methodological rigour.[39]

### Searches and information sources

Searches were carried out in two stages. Stage 1 comprised an initial search in MEDLINE (OvidSP). The population/problems, exposure/issue, outcome/themes framework (see table 1) informed search development. Search concepts were captured using subject headings and text-word searches in Title, Abstract and Keyword Heading Word fields. Search results from stage 1 were assessed by the research team. Following analysis of relevant records, additional terms for inclusion in the search strategy were considered. Further strategy development resulted in a final MEDLINE strategy for use in stage 2 (figure 1). This final strategy was run in MEDLINE (OvidSP) then translated appropriately for a range of databases including: Embase (OvidSP), CINAHL Complete (EBSCOhost), Scopus (www.scopus.com), Social Sciences Citation Index (Web of Science), ProQuest Dissertations & Theses A&I (Proquest), Health Technology Assessment database (https://www.crd.york.ac.uk/CRDWeb), PsycINFO (OvidSP) and Grey Literature Report (http://www.greylit.org/). In addition, the PapersFirst database was searched. The database searches were supplemented by checking the reference lists of included papers. All searches were completed by May 2019.

### Exclusion and inclusion criteria

Papers were selected using specific eligibility criteria outlined in table 2. Only literature focusing on adult death and dying was included. We excluded papers about children, unless they reported on adult children (those aged >18 years) or had a combined focus on adults and children.

### Paper selection

The stage 1 search results were imported into the bibliographical software management package EndNote V.X8.2 and assessed. The stage 2 results were imported into the same EndNote library and results were deduplicated. Stage 2 results remaining after deduplication were assessed. An extensive screening process was undertaken. At the first stage, two independent reviewers (ARo, SL) screened record titles and abstracts for relevance against the screening criteria. Abstracts were double screened (MM) where there were any doubts about eligibility. Full-text papers were screened in pairs (ARo, JT) (MM, SL); both reviewers in each pair independently screened studies for eligibility. A third reviewer resolved eligibility disagreements. Grey literature was reviewed and discussed within the team to agree relevance.

### Data extraction

In line with Arksey and O'Malley's framework data extraction (charting) was multistaged. In stage 1 descriptive characteristics from each included paper were collected. In stage 2 findings and discussion sections of papers were extracted into a data extraction tool. In accordance with Levac *et al*'s[39] recommendation, two reviewers independently extracted and checked data extraction.

### Quality appraisal

While assessing the quality of literature included is not a requirement of scoping reviews, we undertook quality appraisal of included full-text papers from peer-reviewed journals using the Critical Appraisal Skills Programme (CASP) quality assessment tool—a qualitative checklist[40] independently conducted by two researchers. CASP identifies 10 core questions but does not define how overall quality scores should be defined. We scored papers out of 10 and expressed a percentage, those scoring ≥80% were rated as high (H), papers between 60% and 80% as medium (M) and those rated below ≤60% as low (L) (see table 3). Quality appraisal was primarily conducted to illuminate transparency of design, aims and objectives, and

Figure 1   Search strategy.

sample population. However, as is standard to scoping reviews no papers were excluded on grounds of quality. Overall, we scored 29 papers as high quality, 6 medium and 3 low quality. Mixed methods papers with qualitative free-text responses only, conference abstracts, dissertations and book chapters (n=13) were not assessed for quality.

## Data analysis

Data extracted were treated as qualitative data and subject to thematic analysis using Braun and Clarke's approach.[41] Themes were generated during the full-text review and these were discussed within the review team and grouped together. In line with Arksey and O'Malley's framework for analysis a descriptive overview of findings, rather than a full synthesis of the evidence, is provided.[38]

## RESULTS

Searches identified 15 352 records. Following deduplication 8186 records remained for assessment. After assessment, 51 papers were included in the review. Figure 2 shows the review process using the Preferred Reporting Items for Systematic Reviews and Meta-Analyses Extension for Scoping Reviews flow diagram.[35 40]

## Characteristics of papers

Details of included papers are outlined in table 3. Key descriptive information of included papers is as follows: there were 42 journal articles, 5 conference abstracts, 1 book chapter and 3 dissertations. Reported studies were conducted in the UK (n=16), Europe (n=12), USA (n=9), Canada (n=9), Australia (n=3) and New Zealand (n=2). The majority of papers focused on cardiac events (n=15),

**Table 2** Eligibility criteria

| Inclusion criteria | Exclusion criteria |
| --- | --- |
| Papers reporting studies of adult death and dying in prehospital settings | Papers reporting studies of death and dying in healthcare systems outside of Europe, Australia, USA, Canada and New Zealand |
| Papers reporting studies of death and dying as a result of trauma, suicide and/or homicide | Papers reporting studies on response to incidents of death and dying by non-medical emergency services |
| Papers reporting studies of death and dying as a result of natural causes | Papers reporting clinical trials and randomised controlled trials, cohort studies, mixed methods studies without a substantial qualitative element, cost studies |
| Papers reporting studies including families' and/or bystanders' experience of death and dying of a patient where prehospital services respond | Non-English language papers |
| Papers reporting studies of healthcare professionals' experience of providing prehospital care to those who are dying or die | Papers published before 1 January 2000 |
| Papers published in English language | Purely anecdotal or commentary, newspaper articles |
| Papers published between 2000 and 2019 | Papers reporting studies focused on children |
| Qualitative and mixed methods studies (with a substantial qualitative element) | Papers reporting on studies focused on war or terrorism |
| Published conference abstracts/papers | |
| Relevant grey literature from searches (eg, experiences of real clinical practice) | |
| Dissertations and theses | |

and palliative and end-of-life care (EoLC) experiences in PHC (n=21). Suicide (n=4), critical incidents (n=3) and the impact of this work (n=8) were also a focus. Papers reported on the experiences of PHPs only (n=39), families (n=2), families and PHPs (n=6), bystanders (n=2), bystanders and PHPs (n=1) and bystanders and families (n=1). In terms of methodology 38 papers were qualitative and 13 were mixed methods.

Analysis identified four main themes: experiences of death and dying and its impact; experiences of education and training and unmet needs; support needs and experiences; communication and behaviour. Relevant themes are discussed in relation to the three main stakeholder groups: PHPs, families and bystanders.

### PHPs' experiences, education and support needs
#### Experience and impact of death and dying
Experiences of PHPs were characterised by feelings of responsibility towards patients and families and prioritising their needs above their own. There were a number of pressures and stressful tasks identified while attending scenes where dying or death had occurred, including responding alone.[42] Concerns about legal issues related to resuscitations and ethical dilemmas faced by paramedics were also articulated.[43 44] For example, where a do-not-attempt-cardiopulmonary-resuscitation (DNACPR) order was in place, but families requested resuscitation efforts. When providing care to patients, and supporting families and bystanders dealing with death and dying, the emotional labour (the process by which workers manage or suppress

their feelings to maintain an outward appearance to protect or care for the feelings of others)[45] was implicit[46–48] and could result in symptoms of stress and post-traumatic stress disorder.[49] PHPs also described adverse physiological impacts (eg, on sleep, diet) and an impact on family life. Feelings of failure and guilt and thoughts about what had happened to patients after they reached the hospital emergency department were a concern. Critical incidents, such as premature deaths,[50–52] suicide,[46 52–54] OHCA and failed resuscitations,[55] were especially characterised by intense emotional labour and feelings of inadequacy.

In the UK, USA and Canada an increasing demand for PHPs to support patients at EoL was reported. Papers focused on PHPs' perceptions of providing EoLC,[43 56–58] EoL calls,[21 59 60] hospital transfer/transport[56 61 62] and EoLC setting transitions.[62 63] Decision-making about keeping patients close to the EoL at home was complex and experiences of EoLC provision were characterised by emotional labour,[64] care crises, lack of coordination, a need for mediation between services[59] and gaps in communication.[57 59 60 65] PHPs often found it difficult to ascertain patients' EoL wishes. This was compounded by uncertainty and lack of availability of EoL advance directives and care planning, which hindered their ability to keep patients at home.[44 57 61 65] PHPs faced a range of system-level barriers and poor EoLC coordination between services.[62 66] Informing families of death was also noted as an especially stressful, time-consuming and challenging task.[24]

**Table 3** Characteristics of included studies

| Author/year/ publication type | Country and setting | Participants | Aims/objectives | Data collection methods reported | Main findings and conclusions | CASP rating (%), H/M/L* |
|---|---|---|---|---|---|---|
| Brighton et al[64] 2019 Journal paper | UK Hospital, palliative care EoLC | Generalist palliative care staff including ambulance personnel | To explore generalist palliative care providers' experiences of emotional labour when undertaking conversations around palliative and end-of-life care with patients and families, to inform supportive strategies. | Qualitative interviews | Participants reported balancing 'human' and 'professional' expressions of emotion. Support needs included time for emotion management, workplace cultures that normalise emotional experiences, formal emotional support, and palliative and end-of-life care skills training. Diverse strategies to support the emotional needs of generalist staff are crucial to ensure high-quality EoLC and communication, and to support staff well-being. | 10 (100%)—H |
| Carter et al[83] 2019 Journal paper | Canada Community palliative care | Paramedics Family members | To evaluate patient/ family satisfaction and paramedic comfort and confidence following a paramedics in palliative care training programme. | Mixed methods survey interviews | Paramedics describe palliative care as an important and rewarding part of their work. The programme resulted in high patient/family satisfaction and a positive experience of care. Families particularly noted the compassion and professionalism of paramedics. | 8 (80%)—H |
| Fallat et al[77] 2019 Journal paper | USA Prehospital (OOH) | EMS staff Family members | To understand how family members view the ways emergency medical services (EMS) and other first responders interact with distressed family members during an intervention involving a recent or impending paediatric or adult child death. | Mixed methods interviews survey | Family reactions to the crisis and the professional response by first responders were critical to family coping and getting necessary support. Critical competencies identified to help the family cope including: (1) that first responders provide excellent and expeditious care with seamless coordination, (2) allowing family to witness the resuscitation including the attempts to save the child's life, (3) providing ongoing communication. | 5 (50%)—L |

Continued

**Table 3** Continued

| Author/year/ publication type | Country and setting | Participants | Aims/objectives | Data collection methods reported | Main findings and conclusions | CASP rating (%), H/M/L* |
|---|---|---|---|---|---|---|
| Moffat et al[73] 2019 Journal paper | UK Prehospital | Ambulance personnel | To investigate ambulance clinicians' experiences of DNACPR documentation and views concerning potential future changes. | Mixed methods interviews Online questionnaire | Significant increase in numbers of community DNACPR forms has occurred in recent years. Lack of formal DNACPR education, inappropriate CPR attempts and poor communication among stakeholders. Recommendations for a national approach to DNACPR decisions and their documentation. | 9 (90%)—H |
| Ortega-Galán Ángela et al[68] 2019 Journal paper | Spain Hospital, primary care Healthcare centres | Family members | To discover the experiences of end-of-life patients attended by the emergency services, through the discourse of the family caregivers who accompanied the family member in this care transit. | Qualitative interviews Focus groups | Deficiencies in urgent care identified: disorganisation of the care received, lack of experience of the professionals in emergencies, application of general protocols in the emergency services, inadequate care in the treatment received, delays in emergency care. | 8 (80%)—H |
| Waldrop et al[60] 2019 Journal paper | USA Prehospital | Emergency medical technicians Paramedics | To explore prehospital providers' perspectives on how the awareness of dying and documentation of end-of-life wishes influence decision-making on emergency calls near the end of life. | Qualitative interviews | Findings illustrate the relationship between awareness of dying and documentation of wishes in EMS calls. EMS providers are acutely aware of the impact of their decisions and actions on families at the end of life. | 10 (100%)—H |

Continued

**Table 3** Continued

| Author/year/ publication type | Country and setting | Participants | Aims/objectives | Data collection methods reported | Main findings and conclusions | CASP rating (%), H/M/L* |
|---|---|---|---|---|---|---|
| Anderson et al[67] 2018 Journal paper | New Zealand Prehospital | Ambulance personnel | To explore ambulance personnel's decisions to commence, continue, withhold or terminate resuscitation efforts for patients with out-of-hospital cardiac arrest. | Qualitative interviews | Participants sought and integrated numerous factors, beyond established prognostic indicators: prearrival impressions, immediate on-scene impressions, piecing together the big picture and transition to termination of resuscitation. Ambulance personnel may benefit from greater educational preparation and mentoring in managing the scene of a death to avoid inappropriate or prolonged resuscitation efforts. | 10 (100%)—H |
| Donnelly et al[57] 2015 Journal paper | USA Hospice | Emergency medical technicians | To assess the knowledge, attitudes and experiences of EMS providers in the care of patients enrolled in hospice care. | Mixed methods survey including free-text boxes | Themes were family-related challenges, and the need for more education. | Not completed— free-text questionnaire responses only |
| Dow[49] 2018 Dissertation | USA Prehospital | Paramedics Other emergency staff | To look at the relationship between personal, environmental and organisational stress in EMS. | Qualitative interviews Focus groups Observations | Findings signify a need to develop and use stress management and prevention programmes to educate paramedics to increase awareness, recognise the signs and symptoms of stress and learn coping techniques to mitigate the effects encountered. | Not completed— dissertation |

Continued

**Table 3** Continued

| Author/year/ publication type | Country and setting | Participants | Aims/objectives | Data collection methods reported | Main findings and conclusions | CASP rating (%), H/M/L* |
|---|---|---|---|---|---|---|
| Hoare et al[61] 2018 Journal paper | UK Prehospital | Ambulance staff Next of kin | To understand the role of ambulance staff in the admission to hospital of patients close to the end of life. | Qualitative interviews | Ambulance staff have an important role in the admission of end-of-life patients to hospital, frequently having to decide whether to leave a patient at home or to instigate transfer to hospital. Their difficulty in facilitating non-hospital care at the end of life challenges the negative view of near end-of-life hospital admissions as failures. Hospital provision was sought for dying patients in need of care which was inaccessible in the community. | 10 (100%)—H |
| Mainds and Jones[69] 2018 Journal paper | UK Prehospital | Paramedics | To provide an insight into the non-clinical challenges of an OHCA and, how the family members are managed during these difficult incidents. | Qualitative focus groups | Paramedics prefer family not to be present during resuscitation. Use distraction and 'warning shots' throughout resuscitation to prepare the family for bad news. Do not feel sufficiently prepared by their paramedic courses in managing family during OHCAs. Learn how to manage family and BBN by watching experienced colleagues. | 10 (100%)—H |
| Waldrop et al[60] 2019 Journal paper | USA Prehospital | Paramedics | To investigate perceptions of emergency calls at EoL in long-term care facilities. | Qualitative interviews | Contributing factors for calls are care crises; dying-related turmoil; staffing ratios; and organisational protocols. Prehospital providers become mediators between NHS and emergency departments by managing tension, conflict and challenges in patient care between these systems. | 10 (100%)—H |
| Wilson and Birch[63] 2018 Journal paper | Canada Hospital and community settings | Nurses, healthcare professionals, patients and families | To identify current issues and problems with care setting transitions at EoL- producing solutions. | Qualitative interviews | Three inter-related themes were revealed: (A) communication complexities, (B) care planning and coordination gaps, and (C) health system reform needs. | 8 (80%)—H |

Continued

**Table 3** Continued

| Author/year/ publication type | Country and setting | Participants | Aims/objectives | Data collection methods reported | Main findings and conclusions | CASP rating (%), H/M/L* |
|---|---|---|---|---|---|---|
| Armitage and Jones[75] 2017 Journal paper | UK Prehospital | Paramedics | To explore paramedic attitudes towards DNACPR orders. | Mixed methods questionnaire with free-text boxes | The importance of communication in relation to DNACPR orders, as well as the role of allied health professionals and family members in the process. Respecting the patient's wishes was considered paramount, as was educational provision surrounding DNACPRs. | 5 (50%)—L |
| Fernández-Aedo et al[55] 2017 Journal paper | Spain Prehospital | Emergency nurses Emergency medical technicians | To explore the experiences, emotions and coping skills among emergency medical technicians and emergency nurses after performing out-of-hospital cardiopulmonary resuscitation manoeuvres resulting in death. | Qualitative interviews Focus groups | Failed resuscitation results in short and long-term reactions. Negatives, such as sadness or uncertainty, or positives, such as the feeling of having done everything possible to save the patient's life. Emotional stress increases when ambulance staff have to talk with the family of the deceased or when the patient is a child. The workers do not know of a coping strategy other than talking about their emotions with their colleagues. | 10 (100%)—H |
| Kirk et al[43] 2017 Journal paper | UK Prehospital | Paramedics | To understand the perceptions and confidence of paramedics in their role in EoLC in the community. | Survey Open questions text boxes | Paramedics agree EoLC is part of their role but feel they need more education. Length of experience and EoL experience increased confidence. Concerns reported about documentation, litigation and a perceived lack of communication. | Not completed— free-text questionnaire responses only |
| Nilsson et al[46] 2017 Journal paper | Sweden Workplaces | EMS personnel | To describe experiences of supporting survivors of suicide victims from the perspectives of EMS personnel, police officers and general practitioners. | Qualitative focus groups | Professionals make a deliberate choice to acknowledge the needs of survivors by facing their caring responsibilities and providing compassionate care. | 10 (100%)—H |

Continued

**Table 3** Continued

| Author/year/ publication type | Country and setting | Participants | Aims/objectives | Data collection methods reported | Main findings and conclusions | CASP rating (%), H/M/L* |
|---|---|---|---|---|---|---|
| Clompus and Albarran[42] 2016 Journal paper | UK Study centre | Paramedics Emergency care practitioners | To explore how paramedics survive their work within the current healthcare climate. | Qualitative narrative interviews | Coping and resilience was impacted upon via formal methods of support including management, debriefing and referral to outside agencies. Informal methods included peer support, support from family and friends and the use of humour. | 9 (90%)—H |
| Davey et al[44] 2016 Journal paper | New Zealand Prehospital | Paramedics | To highlight and explore underlying values present within practice-based decisions that focus on advance directives. | Survey Free-text responses | Findings revealed legal tensions, multiple constructs of dignity and seeking solutions that support clinical practice. Greater legal guidance and increased professional education in law and ethics are recommended. | Not completed—free-text questionnaire responses only |
| Mathiesen et al[85] 2016 Journal paper | Norway Prehospital | Lay rescuers (bystanders) | To explore lay rescuers' (bystanders) reactions, coping strategies after providing CPR to OHCA victims. | Qualitative interviews | Lay rescuers (bystanders) experience emotional and social challenges, concern and uncertainty after providing CPR in OHCA incidents. Common coping strategies are attempts to reduce uncertainty towards patient outcome and own CPR quality. | 9 (90%)—H |
| Murphy-Jones and Timmons[62] 2016 Journal paper | UK NHS ambulance trust | Paramedics | To explore paramedic decision-making when transporting nursing home residents nearing EoL. | Qualitative interviews | Paramedics identified difficulties in understanding nursing home residents' wishes. Used best interest decision-making, weighing the risks and benefits of hospitalisation. Decision-making became a process of negotiation when the patient's perceived best interests conflicted with that of others, resulting in contrasting approaches by paramedics. | 8 (80%)—H |

Continued

**Table 3** Continued

| Author/year/ publication type | Country and setting | Participants | Aims/objectives | Data collection methods reported | Main findings and conclusions | CASP rating (%), H/M/L* |
|---|---|---|---|---|---|---|
| Peters et al[84] 2016 Journal paper | Australia Prehospital | Bereaved family members following a suicide | To explore participants' perceptions of helpful/ unhelpful interactions with services, family and friends after a suicide death of a family member. | Qualitative narrative | Responses by agencies are often insensitive and not aligned with the needs of those bereaved. Training for agency staff in supporting the suicide bereaved in both the immediate aftermath of a death and their longer term needs is required. | 6 (60%)—L |
| Waldrop and McGinley[66] 2016 Conference abstract | USA Prehospital ambulance care | Prehospital providers | To explore prehospital providers' decision-making when encountering imminent death from serious illness. | Mixed methods survey interviews | EoL challenges in long-term care (LTC) include limited understanding, inconsistent reliance on and variable trust in written directives by LTC staff. EMS providers' decision-making can be solidified by accurate and available written directives. | Conference abstract—not completed |
| Wines[53] 2016 Dissertation | USA Prehospital | Paramedics Emergency medical technicians | To explore paramedics/ emergency medical technicians' experiences responding to completed suicides where the loved one of the deceased is present. | Qualitative interviews | EMS personnel identified experiences of direct and indirect traumatisation as a result of their work. Negative emotions that relate to symptoms of burn-out, compassion fatigue and vicarious traumatisation. Also personal characteristics that mitigate the negative emotions and help them to find meaning in their job. | Dissertation—not completed |
| Hitt 2015[100] Conference abstract | UK Prehospital | Ambulance service Resource dispatchers (RD) | To understand factors influencing RDs' decision-making process when managing ambulance resources attending OHCA and how these decisions might impact on resource availability. | Qualitative interviews | OHCA is prioritised above other time-critical emergencies. Decisions are made rapidly, under pressure and with very little clinical information to hand. A significant amount of time was spent dealing with deceased patients which may affect resource availability and subsequently delay treatment of other critically ill and injured patients. | Conference abstract—not completed |

Continued

**Table 3** Continued

| Author/year/ publication type | Country and setting | Participants | Aims/objectives | Data collection methods reported | Main findings and conclusions | CASP rating (%), H/M/L* |
|---|---|---|---|---|---|---|
| Masquelier et al[76] 2015 Conference abstract | Belgium Hospital | Emergency care provider Family members | To explore how family members and emergency care providers (ECP) perceive and experience family presence during resuscitative events (FPDR) in adult emergency care settings. Also to understand how these perceptions influence their notion of FPDR. | Qualitative interviews | Absolute focus on the patient is of paramount importance. By transferring their needs and perceptions to the background, family members help the ECPs to focus on the patient. In case of a non-successful resuscitation family members and ECP's can reassure each other that all efforts were not in vain. FPDR is for family members an aid in processing the loss of the patient. | Conference abstract—not completed |
| Muller and van der Giessen[78] 2015 Book chapter | Netherlands Emergency medical services | Paramedics Nurses | To describe how violence is dealt with in daily paramedic professional activities. | Qualitative interviews | Paramedics initially ignore verbal abuse because they value the well-being of the patient above their own emotional needs. Managing their own emotions as well as others is essential and achieved through compassion and professionalism—so that bystanders feel that the patient is in good hands. | Book chapter— not completed |
| Rogers et al[79] 2015 Journal paper | Australia Prehospital | St John Ambulance Paramedics | To measure paramedics' perspectives and educational needs regarding palliative care provision, as well as their understanding of the common causes of death. | Mixed methods survey free-text boxes | Paramedics considered palliative care to be focused strongly on EoLC, symptom control and holistic care. The dominant educational needs identified were ethical issues, end-of-life communication and the use of structured patient care pathways. | Not completed – free-text questionnaire responses only |
| Waldrop et al[21] 2015 Journal paper | USA Prehospital | Prehospital providers | To explore and describe prehospital providers' assessments and management of EoL emergency calls. | Qualitative interviews | The importance of managing symptom crises and stress responses that accompany the dying process is essential to quality care at EoL including managing the emotionality of the event and supporting families. | 10 (100%) – H |

Continued

**Table 3** Continued

| Author/year/ publication type | Country and setting | Participants | Aims/objectives | Data collection methods reported | Main findings and conclusions | CASP rating (%), H/M/L* |
|---|---|---|---|---|---|---|
| Jensen et al[80] 2014 Journal paper | Canada Emergency care practitioners (ECP) | Emergency care practitioners | To identify insights gained, lessons learnt from implementation, operation of a novel paramedic long-term care programme. | Qualitative focus groups | The ECP programme has positive implications for the relationship between EMS and LTC, requires additional paramedic training and can positively affect LTC patient experiences during acute medical events. ECPs have a role to play in end-of-life care and find this rewarding. | 9 (90%)—H |
| Munday et al[56] 2014 Conference abstract | UK Prehospital | Paramedics | To understand paramedics' experiences managing patients with advanced cancer and chronic obstructive pulmonary disease (COPD). | Qualitative interviews | Paramedics report managing patients with advanced COPD and cancer to be challenging. However, after undertaking training and receiving support from community professionals, they are able to make decisions to not transfer to ED. Making alternative arrangements was more time consuming than admitting patients to ED. | Conference abstract—not completed |
| Rant and Bregar[54] 2014 Journal paper | Slovenia Emergency medical units | Paramedics Nurses | To understand paramedic nurses' experience of and attitudes to suicidal patients when treating them. | Qualitative interviews | Paramedics demonstrate a professional and understanding approach. They may experience dilemmas while treating suicidal patients, especially those who refuse help or are aggressive. They act according to their subjective risk assessment and previous work experience, yet they lack the expertise to work with suicidal patients, particularly communication skills. | 9 (90%)—H |
| Walker[72] 2014 Journal paper | UK Ambulance trust | Paramedics and nurses | To explore the lived experience of lay presence during adult CPR: out of hospital and in hospital. | Qualitative interviews | There was a combination of benefits and concerns. Familiarity of working in the presence of lay people, practical experience in emergency care and personal confidence were important. Divergent practices within and across the contexts of care were revealed. | 10 (100%)—H |

Continued

**Table 3** Continued

| Author/year/ publication type | Country and setting | Participants | Aims/objectives | Data collection methods reported | Main findings and conclusions | CASP rating (%), H/M/L* |
|---|---|---|---|---|---|---|
| Douglas et al[24] 2013 Journal paper | Canada Paramedic service | Paramedics | To explore paramedics' experiences with death notification education. | Qualitative focus groups | Paramedics learn to communicate death notifications by observing others and by trial and error and there is a lack of formal death notification education. Paramedics want to learn about the practical aspects of communicating death notifications, managing the reactions of the bereaved, the cultural and religious aspects of death, as well as their personal reactions to death. | 8 (80%)—H |
| Møller et al[86] 2013 Conference abstract | Denmark Prehospital | Medical dispatchers Lay people (bystanders) | To develop a concept for systematic feedback to lay people by exploring lay peoples' need for feedback interviews after performing CPR and by identifying practical and legal barriers to provide systematic feedback. | Qualitative interviews | Themes identified were the challenge of identifying OHCA, collaboration with the medical dispatcher and the ambulance crew, coping with the experience of sudden death, reflections on what more could have been done and experience for the future, the outcome of the patient and the perceptual experience with OHCA. | Conference abstract—not completed |
| Robinson et al[65] 2013 Journal paper | UK Prehospital | Ambulance service workers Legal professionals | To explore professionals' experiences on the implementation of advance care planning in two areas of clinical care: dementia and palliative care. | Qualitative focus groups Interviews | There was uncertainty over the general value of advance care planning, whether current service provision could meet patient wishes, their individual roles and responsibilities and which aspects of advance care planning were legally binding; the array of different advance care planning forms and documentation available added to the confusion. | 10 (100%)—H |
| Williams[47] 2013 Journal paper | UK Paramedic students | Preregistration Paramedic science students | To explore student paramedic perceptions and experiences of emotion work and the strategies used to deal with it. | Qualitative interviews | The findings reveal evidence of emotion work in emergency situations where there is a need to control and suppress emotions to do the job, struggling with emotion and a need for talking it through. | 7 (70%)—M |

Continued

**Table 3** Continued

| Author/year/ publication type | Country and setting | Participants | Aims/objectives | Data collection methods reported | Main findings and conclusions | CASP rating (%), H/M/L* |
|---|---|---|---|---|---|---|
| Bremer et al[71] 2012 Journal paper | Sweden Prehospital | EMS personnel | To analyse EMS personnel's experiences of caring for families when patients suffer from cardiac arrest and sudden death. | Qualitative interviews | EMS felt responsible for both patient and family care, and sometimes failed to prioritise these responsibilities as a result of their own perceptions, feelings and reactions. Moving from patient care to family care implied a movement from well-structured guidance to a situational response, where the personnel were forced to balance between interpretive reasoning and a more direct emotional response. Ethical caring competence is needed in the care of bereaved family members to avoid additional suffering. | 9 (90%)—H |
| Douglas et al[25] 2012 Journal paper | Canada Prehospital | Ambulance service Paramedics Primary care Advance care | To explore paramedics' experiences and coping strategies with death notification in the field. | Qualitative focus groups | Paramedics' experiences with death notification are stressful, challenging and rewarding. More formal support for paramedics is necessary, especially when the nature of the death is distressing. | 6 (60%)—L |
| Lord et al[58] 2012 Journal paper | Australia Prehospital | Paramedics | To identify paramedics' knowledge, beliefs and attitudes related to the care of patients requiring palliative care in community health settings. | Qualitative focus groups Interviews | Findings identified conflict in goals of care, legal issues, access to information and challenges of organisational policy and clinical practice guidelines. | 7 (70%)—M |
| Timmons et al[87] 2010 Journal paper | UK Public places | Staff trained in first aid/AED use working in public places | To explore perceptions of the training how staff understood the use of the automated external defibrillator. | Qualitative interviews | The interpreted social affordance of the AED was to delay and displace the moment and site of death and confirms that death in public space is a disturbing event for those involved in dealing with the death and its aftermath. | 6 (60%)—L |

Continued

**Table 3** Continued

| Author/year/ publication type | Country and setting | Participants | Aims/objectives | Data collection methods reported | Main findings and conclusions | CASP rating (%), H/M/L* |
|---|---|---|---|---|---|---|
| Bremer et al[70] 2009 Journal paper | Sweden Prehospital | Families Emergency personnel | To describe the experiences of significant others present at OHCA, focusing on ethical aspects and values. | Qualitative interviews | OHCA can be stated as unreality in the reality and is characterised by overwhelming responsibility. The significant others experience inadequacy and limitation, they move between hope and hopelessness and they struggle with ethical considerations and an insecurity about the future. | 10 (100%)—H |
| Halpern et al[50] 2009a Journal paper | Canada EMS organisation | Mandatory continuing medical education programme volunteers | To characterise critical incidents and elicit intervention suggestions. | Qualitative interviews Focus groups | Ambulance workers suffer considerable distress from critical incidents and would welcome interventions. Difficulty in acknowledging distress and fear of stigma presented significant barriers to accessing support. | 10 (100%)—H |
| Halpern et al[51] 2009b Journal paper | Canada EMS organisation | Emergency medical technicians | To explore and describe emergency medical technicians' (EMTs) experiences of critical incidents and views about potential interventions, in order to facilitate development of interventions that take into account EMS culture. | Qualitative interviews Focus groups | Following critical incidents ambulance workers identify two workplace resources in the immediate aftermath of an incident: supervisor support; and a brief time out period in which to talk informally, often with peers as important for their recovery. | 9 (90%)—H |
| Gallagher and McGilloway[52] 2008 Journal paper | UK/Ireland Ambulance care | Emergency medical technicians Emergency medical clinicians | To assess the nature and impact of critical incidents on health and well-being; examine attitudes towards support services; and explore barriers to service use. | Qualitative interviews | Exposure to critical incidents has a significant impact on health and well-being; this has important implications for recognising and appropriately addressing the health and training needs of ambulance personnel, including the effective management of critical incident stress. | 9 (90%)—H |

Continued

**Table 3** Continued

| Author/year/ publication type | Country and setting | Participants | Aims/objectives | Data collection methods reported | Main findings and conclusions | CASP rating (%), H/M/L* |
|---|---|---|---|---|---|---|
| Andrus[74] 2007 Dissertation | USA Community hospital | Volunteer first aid squads Volunteer EMTs | To explore volunteer EMTs' understanding of out-of-hours DNR. | Mixed methods survey Narrative interviews | Findings indicate a lack of out-of-hospital do-not-resuscitate orders at cardiac arrest calls; benefits and harms of cardiopulmonary resuscitation; chaotic cardiopulmonary resuscitation and family environments and EMTs as virtuous agents. There are also ethical versus legal concerns and potential for getting drawn into drama of family tragedy. | Dissertation—not completed |
| Jonsson and Segesten[82] 2004 Journal paper | Sweden Ambulance stations | Ambulance staff | To uncover and obtain in-depth understanding of the way ambulance staff experience and handle traumatic events and to develop an understanding of the life world of the participants. | Qualitative interviews | The findings show that post-traumatic stress symptoms, guilt, shame and self-reproach are common after duty-related traumatic events. To handle these overwhelming feelings it is necessary to talk about them with fellow workers, friends or family members. | 8 (80%)—H |
| Jonsson and Segesten[81] 2003 Journal paper | Sweden Ambulance care | Ambulance workers (nurses) | To uncover the essence of traumatic events experienced by Swedish ambulance personnel. | Qualitative written stories | Findings indicate that staff have a strong identification with the victims and it is impossible to prepare for events that are unforeseen and meaningless. To handle the overwhelming feelings of identification, ambulance personnel have to gain understanding through talking about those feelings. | 9 (90%)—H |
| Regehr 2003[101] Journal paper | Canada Emergency service | Emergency service Professionals Paramedics Firefighters Police | To understand experiences when testifying at postmortem reviews following death of person in their care, death during involvement in incident. | Qualitative Method not reported | To meet their goal of improving service, it is important that organisations provide support for emergency responders participating in death inquiries. | 6 (60%)—L |

Continued

**Table 3** Continued

| Author/year/ publication type | Country and setting | Participants | Aims/objectives | Data collection methods reported | Main findings and conclusions | CASP rating (%), H/M/L* |
|---|---|---|---|---|---|---|
| Regehr et al[48] 2002 Journal paper | Canada Ambulance care Emergency service | Paramedics | To better understand factors that lead to higher levels of distress among paramedics. | Mixed methods Questionnaires Interviews | Paramedics deal with the events cognitively and technically while maintaining an emotional distance. At times, an emotional connection with events based on their awareness of other aspects of the patient's experience. When this occurs, paramedics report increased symptoms of traumatic stress. | 8 (80%)—H |
| Ruston[88] 2001 Journal paper | UK General hospitals | Patients Relatives/bystanders | To explore lay decision-making at the time of a cardiac event and address the question of why people do not call for an ambulance. | Qualitative interviews | Lack of knowledge of the role of emergency services and confusion about whether symptoms were serious enough to warrant calling for an ambulance. | 3 (30%)—L |

*Denotes rating of high (H), medium (M) and low (L).

AED, automated external defibrillation; BBN, breaking bad news; CASP, Critical Appraisal Skills Programme; CPR, cardiopulmonary resuscitation; DNACPR, do-not-attempt-cardiopulmonary-resuscitation order; DNR, do-not-resuscitate order; ED, emergency department; EoL, end of life; EoLC, end-of-life care; NHS, National Health Service; OHCA, out-of-hospital cardiac arrest; OOH, out of hospital.

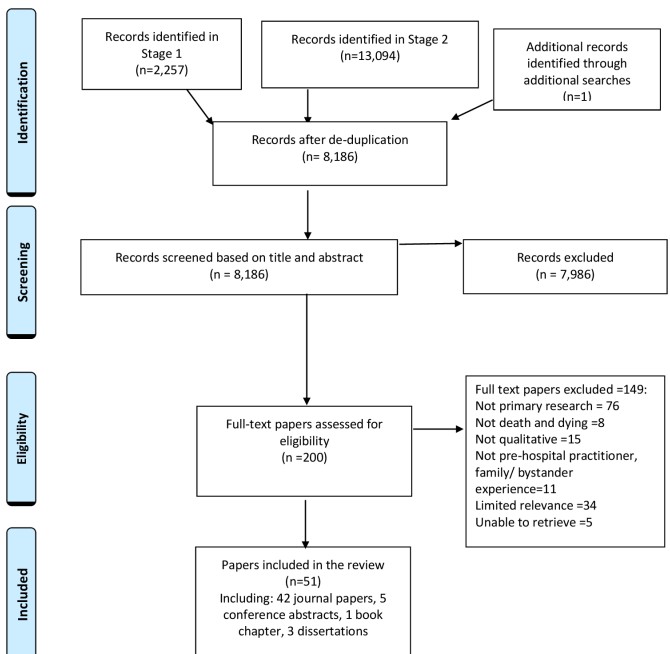

**Figure 2** Preferred Reporting Items for Systematic Reviews and Meta-Analyses Extension for Scoping Reviews (PRISMA-ScR) flow diagram.[35]

Resuscitation and OHCA were also described as particularly intense and stressful experiences, characterised by intense emotions.[55 67–71] PHPs described complex decision-making around cardiac arrests,[65] the technical abilities required alongside providing holistic care[72] and difficult processes of negotiation with coworkers, other HCPs and families. There was inadequate communication around out-of-hospital DNACPR orders.[73–75]

## Managing the work of death and dying
PHPs used a variety of strategies to manage the work of caring for patients who died or were dying, such as collecting as much on-scene information as possible before arrival, especially when attending OHCAs.[67] This was important for understanding the context and dealing with emotional aspects. While on scene, coping strategies such as detachment, surface acting,[42 53] humour[42] and, in the case of suicides, focusing on survivors were used. Despite the psychological and physiological impact of caring for patients and relatives in prehospital contexts of death and dying PHPs also identified personal meaning, such as identifying with families because of their own experience of death and rewards from work in this environment[53] including a sense of pride from a job well done.[76]

## Perceptions on the presence of families
The presence of family members and bystanders when attending death and dying calls provoked mixed reactions from PHPs, irrespective of context (eg, OHCA, EoLC).[55 71] During OHCAs, some PHPs preferred families not to be present in the room during resuscitation and described feeling pressure from families, and expressions

of disquiet related to their presence.[69] Others considered that relatives could aid resuscitation by putting on hold their own feelings and emotions, thus helping PHPs focus on the condition of the patient.[76 77] PHPs and families could provide mutual reassurance following unsuccessful resuscitations, that efforts were not inappropriate.[76] Where there were conflicts with family members, for example, over resuscitation,[76] staff managed these tensions by assigning them roles in the resuscitation efforts, such as giving them a bag of fluids to hold.[78] The feeling that relatives were being useful could help prevent tensions from escalating.

## Training and education needs
A lack of sufficient training around communication and relational aspects of death and dying, palliative and EoLC,[24 25 43 58–60 79 80] OHCA and DNACPR,[67 69 73 75] death notifications[24 25] and breaking bad news[55 69] was evident in the review. PHPs identified deficits in training around DNACPR orders,[73 75] dealing with suicide,[46 52–54] managing bereavement,[71] communicating bad news and emotion work.[47 50 69] PHPs attending OHCAs felt training did not adequately prepare them to manage families during cardiac arrests.[69] In addition to better training, papers identified a need for national-level guidance and documentation around resuscitation.[73] PHPs described learning on the job through observing their colleagues,[69] and wanted training and mentoring from other HCPs and peers.[67]

## Support needs
In a work environment, characterised by intense emotional demands, PHPs reported a need for several dimensions of support not always available, including time out periods, protected time after stressful calls, and colleague, supervisor and management support.[50 51] A physical space for reflection and collective support from peers was especially important.[50 51] Support received from management and at an organisational level was described as mixed ranging from positive, empathic support and provision of time out,[50 51] to an absence of a climate of care[52] and lack of concern from management.[50 51 81 82] Staff described the stigma surrounding expressions of stress experienced within organisations[50 51] and while professional services and peer support services were available,[52] uptake of these was variable, with concerns raised about being treated as an 'outcast' for accessing such services.

## Experiences, impact and needs of families
Few papers focused on family and significant others' experiences, but those that did reported the lasting impact of these events.[61 68 70 77 83 84] Included papers reported on family experiences in prehospital palliative and EoLC,[61 68 83] OHCA,[70 77] resuscitation[76] and suicide.[84] Witnessing a family member die or dying was reported as having a significant effect on relatives and particularly at OHCAs, families experienced a range of emotions,

including reactions of shock, vulnerability, responsibility and hopelessness.[70]

## Experiences of behaviour and communication

Families described witnessing PHPs exhibiting calm and control in difficult situations, and this included interactions with parents where their adult child had died.[77] In general, family members reported experiencing mainly positive behaviours and communication with PHPs who provided competent care,[77] and in cases involving suicide showed kindness, empathy and compassion.[84] In cases where there was a death of adult children, families reported being treated with dignity by PHPs.[77] There were some reports of relatives experiencing negative interactions with PHPs, where they demonstrated a lack of awareness of family-centred practice,[77] insensitivity or little compassion in cases of suicide.[84] In such instances, families felt further training was needed.

## Experiences, impact and needs of bystanders

There was a paucity of evidence around experiences and perspectives of bystanders. Papers which discussed bystander experiences, described difficulties associated with cardiac events,[85–88] including identifying OHCAs.[79] Bystanders reported a lack of knowledge around emergency services, confusion over patient symptoms requiring ambulance response at the time of cardiac events and differing opinions on actions needed and when to call for an ambulance.[88] While the literature is limited, it appeared that irrespective of whether bystanders were passers-by or present at events, they still experienced ongoing adverse reactions. These included social and psychological disturbance (eg, guilt, self-criticism) following witnessing deaths or giving cardiopulmonary resuscitation (CPR) and automated external defibrillation (AED) at cardiac events.[85–87] Møller et al described this as 'the perceptual OHCA experience' whereby bystanders ruminate 'on what more could have been done' (p S22).[86] Being health educated was considered to offer some mitigation against these concerns.[85] In the UK, those working in public places and trained to use AEDs for OHCA also reported negative consequences including flashbacks.[87] There was an identifiable need on the part of bystanders to witness visible resuscitation efforts on the part of emergency services, for feedback following sudden deaths[86] and information on patient outcomes.[85]

## DISCUSSION

We conducted a scoping review to identify and explore factors that characterise and shape PHP, family and bystander experience of death and dying in PHC and identify gaps in knowledge that warrant further research. The review identified a developing evidence base on PHPs' experiences, particularly in the UK, Europe and North America. However, there were significant shortcomings in the literature in regard to the experiences,

needs and impact of death and dying for families and bystanders.

Our review confirmed existing research of the varied and complex work of PHPs,[15] often requiring them to respond to a range of time-critical emergencies including cardiac events,[89] placing them in situations that could be difficult to manage and which presented a range of challenges and emotional demands. This was the case for PHPs with variable expertise or length of experience and often required they used a variety of coping strategies. While PHPs reported feeling confident to undertake the clinical elements of managing a patient who was dying or who died, they often felt less prepared for handling the more emotional aspects particularly when it involved communicating bad news.

This was also the case for providing palliative and EoLC, which is an increasing part of PHP's role, particularly in the UK, USA and Canada. In the UK, issues surrounding quality of access to EoLC services and the reorganisation of ambulance services to provide support to patients at the EoL may in part explain this growth in PHC EoLC provision.[90] The multiple challenges that faced PHPs attending EoLC calls often meant having to use skills of crisis and conflict management and carry out the emotional support work of death and dying for which they reported minimal preparation or training. This places additional pressures on PHPs, already faced with complex decision-making and the complexities of providing care to patients nearing the EoL, and whose actions and handling of these situations influences how people die and whether their preferences are respected. Similarly, for families, given that they may not have experienced death or dying previously, how this work is managed by PHPs is likely to influence the transition to bereavement. In the UK, the key role of PHPs in the care of those at the EoL is recognised in policy,[91] and guidance on delivering EoLC[92 93] and breaking bad news[94] has been developed and is now reflected in their training curriculum and includes preparation for the moral work they will need to engage in as a result of the emotional challenges they are likely to encounter.[95] However, the experiences identified in this review suggest that challenges remain in the application of these recommendations and training in the real-world setting of PHC. Therefore, further research is needed to understand if, for example, these challenges are a result of stress resulting from the incident or coping mechanisms, rather than inadequate training.

While the review informed our understanding of PHPs who attended at the scene, we identified little qualitative evidence related to understanding the experience or impact of death and dying on call handlers. This is despite them being the first point of contact and managing situations involving death and dying as an integral part of their role. The job of a call handler is stressful and the psychological impact of dealing with emergency calls has been widely documented elsewhere.[96 97] However, currently we know little about the specific impact of dealing with these

aspects on those undertaking this role, the challenges they face, the extent to which these are related to the nature of the role itself or organisational factors, and training and support needs. Given current concerns around the mental health of emergency service workers, and that the need for an evidence base has been highlighted recently by those who support them,[98] this is clearly an omission that merits further investigation.

Papers that included family members' accounts and experiences were few and tended to focus on their interactions with HCPs, including communication and behaviours during resuscitation and cardiac events,[70 76 77] EoLC,[61 68 83] the impact of cardiac events[70 77 88] and occasionally experiences of suicide.[84] Generally, relatives reported positive interactions with PHPs, commenting on their confidence and calmness in attending scenes involving death and dying, and while some families reported more negative encounters, it suggests there may be a disconnect between PHPs' perceptions of the care they provide and families' experience of that care. From the minority of papers identified on bystanders' experiences, there appears to be limited support available to those who have experienced stress or other symptoms from their involvement in events such as resuscitation for OHCA, or discussions about what form such support might take. Identifying and developing support mechanisms for this group will become increasingly important with the move towards encouraging bystander CPR and public access defibrillation which are key determinants in OHCA survival prior to PHP arrival.[99]

A paucity of evidence relating to families and bystanders' experiences and support needs is an important knowledge gap. There may be several explanations for this limited evidence base. For example, undertaking thanatological research with families and bystanders in the PHC context is likely to present both methodological and ethical challenges perceived by researchers as potential barriers to conducting research in this area. Nevertheless, as both participants in, and observers of, death and dying in the prehospital setting, applied research that addresses questions about experiences and impact and subsequently leads to the development of appropriate interventions is essential.

## CONCLUSION

This review has shown there is a broad consistency regarding the experience of PHPs in relation to dealing with death and dying. It also identified current gaps in knowledge and areas where further empirical research that addresses specific research questions is needed. In particular, the limited evidence on call handlers suggests it is imperative to explore whether their experiences and needs are the same as those PHPs who attend at scene, or if there are differences between the two groups that need to be considered. There is also a need to investigate the effectiveness of current training in order to identify if gaps exist and the translation of this knowledge into

practice and how this supports a rapidly evolving service. The paucity of evidence on families and bystanders presents opportunities to investigate their experiences in greater depth so that we can begin to understand their needs and how these can be addressed. Future research to address the current knowledge gaps will be important for informing future policy and practice for managing death and dying in the prehospital context.

**Author affiliations**
[1]School of Health Sciences, University of Southampton, Southampton, UK
[2]York Health Economics Consortium, University of York, York, UK
[3]University Hospital Southampton NHS Foundation Trust, Southampton, UK
[4]South Central Ambulance Service NHS Foundation Trust Southern Headquarters, Otterbourne, UK
[5]Warwick Clinical Trials Unit, Warwick Medical School, University of Warwick, Coventry, UK
[6]NIHR Southampton Respiratory Biomedical Research Unit, University of Southampton, Southampton, UK

**Contributors** MM, SL and ARi designed the review. MA developed the search strategy and performed the searches. SL, ARo, MM and JT screened the titles, abstracts and full papers. SL and ARo performed data extraction. MM, SL, ARo carried out data analysis. MM, SL and ARo drafted the manuscript. MM, SL, ARi, JT, ARo, MA, RC, HP and CD reviewed the paper for important intellectual content. MM, SL, ARi, JT, ARo, MA, RC, HP and CD approved the final version of the paper.

**Funding** This work was supported by the National Institute for Health Research (NIHR) through the Collaboration for Leadership in Applied Health Research and Care Wessex (NIHR CLAHRC Wessex) programme.

**Disclaimer** The views expressed are those of the authors and not necessarily those of the NHS, the NIHR or the Department of Health and Social Care. The funders had no role in study design, data collection and analysis, decision to publish, or preparation of the manuscript.

**Competing interests** Alison Richardson is a National Institute for Health Research (NIHR) senior investigator. HP is in receipt of an NIHR Clinical Doctoral Fellowship.

**Patient consent for publication** Not required.

**Provenance and peer review** Not commissioned; externally peer reviewed.

**Data availability statement** Data sharing not applicable as no data sets generated and/or analysed for this study. No additional data are available.

**ORCID iD**
Michelle Myall http://orcid.org/0000-0001-8733-7412

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
