## [Reviewer comments · BMJ Open]

ARTICLE DETAILS

TITLE (PROVISIONAL)	Death and dying in pre-hospital care: what are the experiences and issues for pre-hospital practitioners, families and bystanders? A scoping review.
AUTHORS	Myall, Michelle; Rowsell, Alison; Lund, Susi; Turnbull, Joanne; Arber, Mick; Crouch, Robert; Pocock, Helen; Deakin, Charles; Richardson, Alison

VERSION 1 – REVIEW

REVIEWER	Birgitta Wireklint Sundström Faculty of Caring Science, Work Life and Social Welfare, University of Borås, SE-501 90 Borås, Sweden
REVIEW RETURNED	17-Feb-2020

GENERAL COMMENTS	1. Importance of submission This is a manuscript of insufficient interest to warrant publication. The strengths are that the highlighted issue – death and dying in pre-hospital care – is of big interest; therefore, the research is demanded and has relevance. This manuscript is generally well written. 2. Theoretical evaluation The manuscript lacks an ethical analysis of/reflection on pre-hospital practitioners' work. It is necessary to expand the INTRODUCTION with some articles on ethical problems within pre-hospital care in the context of death and dying outside of the hospital environment where pre-hospital practitioners respond. This extension fits the paragraph that starts with "The provision of care to someone who is dying, and their relatives, has been identified as one of the most stressful human experiences". Readers need to know what ethical problems might be behind the stressful human experiences. There are also a need to discuss more than strategies and training in relation to their need of emotional support. Thus, some ethical issues should be clearly stated within the DISCUSSION and in the STRENGTHS AND LIMITATIONS. A statement about ethical awareness in the second bullet should be appropriate. 3. Design and methods The design is clear and methods is an ambitious part of the manuscript. In the paragraph "Quality appraisal"; you refer to Table 2 "(See Table 2)". However, this reference appears incorrect. This review is limited to western countries similar to healthcare systems that in UK. However, a literature review conducted evidence from western countries only must be reported as a limitation in DISCUSSION and in STRENGTHS AND LIMITATIONS. 4. Results
---

	In “Characteristics of papers”; you present number of articles from different western countries. Please count the number for Canada and USA, once more. 5. Comments Accept with changes.
--	---

REVIEWER	Natalie Anderson The University of Auckland, New Zealand Auckland City Hospital Emergency Department, New Zealand
REVIEW RETURNED	31-Mar-2020

GENERAL COMMENTS	Thank you for the invitation to review this well-constructed and relevant scoping review paper. My congratulations to the authors, for undertaking and authoring this scoping review. It is particularly timely and topical in light of the COVID19 pandemic. This scoping review identifies and synthesises existing research into death and dying in the pre-hospital setting, with a focus on the experiences of pre-hospital practitioners, families and bystanders. As noted by the researchers, there is limited research into death and dying in the unique, emergency services context. This review helps to identify what is known and where further research is needed. The paper is generally very well-written, and I found most of the text clear and easy to read. Finding a collective term for all the pre-hospital care providers in the included studies presents a challenge. The term 'pre-hospital' is often useful in this domain, but in the context of death and dying, few patients are transported to hospital. Therefore, I'm not sure it best captures the study focus. I wonder if community emergency responders / community emergency response might be better? P3 Lines 34-40 Please revise the third point in the 'strengths and limitations' section, as your meaning is unclear P5 Line 36. You have cited the number of OHCA's 'treated' by ambulance personnel in the UK in 2014. As the focus of this review is care in the context of death and dying, it is equally relevant to note that this number represents only half of the OHCA's attended by ambulance personnel. Although 'treated' is defined as active resuscitation efforts by ambulance personnel, ambulance care was presumably provided in almost all of the 60,000 OHCA call-outs in the form of patient assessment, breaking bad news and supporting family and bystanders. Please complete a PRISMA-ScR [1] checklist and submit this, along with your revisions. Please clearly rationalise your exclusion criteria. Why did you exclude all quantitatively-focussed research? You have acknowledged that models of emergency service provision vary significantly around the world. Why did you exclude research outside Europe, Australia, US, Canada and NZ? How could these exclusions limit your findings? Paper #18 in the reference list appears to have been published in 2017, not 2016. 1. Tricco AC, Lillie E, Zarin W, O'Brien KK, Colquhoun H, Levac D, et al. PRISMA extension for Scoping Reviews (PRISMA-ScR): Checklist and explanation. Ann Intern Med. 2018;169(7):467-473. Available from: https://doi.org/10.7326/M18-0850
--

VERSION 1 – AUTHOR RESPONSE

Response to Reviewers' Comments

Reviewer 1

Comment	Response
This is a manuscript of sufficient interest to warrant publication. The strengths are that the highlighted issue – death and dying in pre-hospital care – is of big interest; therefore, the research is demanded and has relevance. This manuscript is generally well written.	Thank you for your supportive comments.
The manuscript lacks an ethical analysis of/reflection on pre-hospital practitioners' work. It is necessary to expand the INTRODUCTION with some articles on ethical problems within pre-hospital care in the context of death and dying outside of the hospital environment where pre-hospital practitioners respond. This extension fits the paragraph that starts with "The provision of care to someone who is dying, and their relatives, has been identified as one of the most stressful human experiences". Readers need to know what ethical problems might be behind the stressful	This paragraph now includes some examples to illustrate the ethical issues that pre-hospital practitioners face in the course of their work dealing with death and dying, with additional references cited. This section also includes a brief overview of the additional ethical challenges and 'moral work' faced by pre-hospital practitioners in the current global COVID-19 pandemic.

human experiences.	
There are also a need to discuss more than strategies and training in relation to their need of emotional support. Thus, some ethical issues should be clearly stated within the DISCUSSION and in the STRENGTHS AND LIMITATIONS. A statement about ethical awareness in the second bullet should be appropriate.	Thank you for your comment. We consider that the ethical demands of the role in dealing with death and dying are already included but have added to the sentence regarding inclusion of ethical challenges in the paramedic curriculum (page 34, line 339-340). We have added a sentence to the second bullet in the strengths and limitations which points to the need for support and training to include the ethical challenges faced in the work of dealing with death and dying.
The design is clear and methods is an ambitious part of the manuscript.	Thank you for this supportive comment.
In the paragraph "Quality appraisal"; you refer to Table 2 "(See Table 2)". However, this reference appears incorrect.	Thank you for alerting us to this error. We have amended and corrected this reference to Table 3.
This review is limited to western countries similar to healthcare systems that in UK. However, a literature review conducted evidence from western countries only must be	We have added to Bullet Point 4 in Strengths and Limitations to reflect this as a limitation.

reported as a limitation in DISCUSSION and in STRENGTHS AND LIMITATIONS.	
In "Characteristics of papers"; you present number of articles from different western countries. Please count the number for Canada and USA, once more.	Thank you. We have checked this and amended the number of articles included for USA and Canada. This should be 9 papers for both countries.
Accept with changes	Thank you.
Reviewer 2	
Comment	Response
Thank you for the invitation to review this well-constructed and relevant scoping review paper. My congratulations to the authors, for undertaking and authoring this scoping review. It is particularly timely and topical in light of the COVID19 pandemic. This scoping review identifies and synthesises existing research into death and dying in the pre-hospital setting, with a focus on the experiences of pre-hospital practitioners, families and bystanders. As noted by the researchers, there is limited research into death and dying in the unique, emergency services context. This review helps to identify what is known and where further research is needed. The paper is generally very well-written, and I found most of the text clear and easy to read.	Thank you for your supportive comments.
Finding a collective term for all the pre-hospital care providers in the included studies presents a challenge. The term 'pre-hospital' is often useful	We are using this term to primarily describe paramedic / ambulance services as explained in the first paragraph on page 4 (lines 2-7). We

in this domain, but in the context of death and dying, few patients are transported to hospital. Therefore, I'm not sure it best captures the study focus. I wonder if community emergency responders / community emergency response might be better?	agree that paramedics (and other practitioners/first responders) now do a lot on on-scene / non-conveyance work as well as transferring to hospital. We consider the term 'pre-hospital care' captures 'conveyance work' as well as 'in situ' / managing at scene (including death and dying). Pre-hospital care is a more widely understood term (particularly in the UK) community emergency responders doesn't really translate to the UK context where the term is used primarily to refer to volunteers.
P3 Lines 34-40 Please revise the third point in the 'strengths and limitations' section, as your meaning is unclear	We have revised the wording of this third point in the strengths and limitations which hopefully makes the meaning clear.
P5 Line 36. You have cited the number of OHCAs 'treated' by ambulance personnel in the UK in 2014. As the focus of this review is care in the context of death and dying, it is equally relevant	Thank you for this helpful comment. We have added a sentence to make it clear that care was provided in a variety of forms and not just active resuscitation efforts (page 5, lines 38-41)

to note that this number represents only half of the OHCA's attended by ambulance personnel. Although 'treated' is defined as active resuscitation efforts by ambulance personnel, ambulance care was presumably provided in almost all of the 60,000 OHCA call-outs in the form of patient assessment, breaking bad news and supporting family and bystanders.	
Please complete a PRISMA-ScR [1] checklist and submit this, along with your revisions.	Apologies this was an omission from the original submission. The PRISMA-ScR checklist has now been submitted as part of the revisions.
Please clearly rationalise your exclusion criteria. Why did you exclude all quantitatively-focussed research? You have acknowledged that models of emergency service provision vary significantly around the world. Why did you exclude research outside Europe, Australia, US, Canada and NZ? How could these exclusions limit your findings	We did not include papers reporting studies using quantitative methods because we were interested in understanding experience of death and dying in pre-hospital care. Papers reporting studies using methodologies/methods such as clinical trials and randomised controlled trials, cohort studies were considered unlikely to provide an understanding of lived experience of death and dying for healthcare professionals, families or bystanders. Papers reporting studies who employed mixed methods where there was a qualitative component were included. We focused on those countries who have healthcare systems that are similar to the UK but acknowledge this may be a limitation of the review. We have amended Bullet 4 in 'Strengths and Limitations' for reflect this.
Paper #18 in the reference list appears to have been published in 2017, not 2016.	Thank you. We have amended the year of publication to 2017.

VERSION 2 – REVIEW

REVIEWER	Natalie Anderson The University of Auckland, New Zealand
REVIEW RETURNED	22-Jun-2020
GENERAL COMMENTS	My congratulations to the authors, this manuscript has been improved in response to reviewer feedback and provides a timely overview of an important area of prehospital practice.